# Effects of the γ″-Ni_3_Nb Phase on Fatigue Behavior of Nickel-Based 718 Superalloys with Different Heat Treatments

**DOI:** 10.3390/ma12233979

**Published:** 2019-11-30

**Authors:** Li-Shi-Bao Ling, Zheng Yin, Zhi Hu, Jun Wang, Bao-De Sun

**Affiliations:** 1Shanghai Key Lab of Advanced High-temperature Materials and Precision Forming, Shanghai Jiao Tong University, Shanghai 200240, China; zhengyin1114@163.com (Z.Y.); junwang@sjtu.edu.cn (J.W.);; 2Mechanical and Electrical Engineering College, Nanchang University, Nanchang 330031, China; huzhi@ncu.edu.cn

**Keywords:** γ″-Ni_3_Nb, fatigue behavior, nickel-based 718 superalloy, heat treatment, hot isostatic pressing

## Abstract

The effects of the γ″-Ni_3_Nb phase on fatigue behavior of nickel-based 718 superalloys with standard heat treatment, hot isostatic pressing + solution treatment + aging, and hot isostatic pressing + direct aging were investigated by scanning electron microscope, transmission electron microscopy, and fatigue experiments. The standard heat treatment, hot isostatic pressing + solution treatment + aging, and hot isostatic pressing + direct aging resulted in the formation of more and smaller γ″ phases in the matrix in the nickel-based 718 superalloys. However, the grain boundaries of the hot isostatic pressing + direct aging sample showed many relatively coarse disk-like γ″ phases with major axes of ~80 nm and minor axes of ~40 nm. The hot isostatic pressing + direct aging sample with a stress amplitude of 380 MPa showed the longest high cycle fatigue life of 5.16 × 10^5^ cycles. Laves phases and carbide inclusions were observed in the crack initiation zone, and the cracks propagated along the acicular δ phases in the nickel-based 718 superalloys. The precipitation of fine γ″ phases in the matrix and relatively coarse γ″ phases in the grain boundaries of the hot isostatic pressing + direct aging sample can hinder the movement of dislocation.

## 1. Introduction

Nickel-based 718 superalloys, which have excellent mechanical properties, high temperature oxidation resistance, and fatigue resistance at elevated temperatures, have been used extensively in aerospace, gas turbine engines, and power generation for decades [1,2,3,4,5,6]. The ever-increasing demand for more efficient and powerful turbine engines continues to drive the need to improve or develop a new generation of nickel-based 718 superalloys [7]. However, fatigue damage readily occurred in the turbine disc, which was subjected to the combined effects of mechanical stress and high temperature while in service [8]. Additionally, the increase in fatigue damage may lead to ultimate catastrophic failure of the structural component under subsequent cyclic loads [9]. Therefore, it is necessary and significant to improve the fatigue life of the nickel-based 718 superalloys.

Among all the properties of superalloys, fatigue is one of the most important, often limiting the overall service life. Generally, the microstructures, such as segregation, δ phases, γ’ phase, γ″ phase, and carbide particles, have a significant effect on fatigue behavior of nickel-based 718 superalloys [10,11,12,13]. Zhang et al. [14] indicated that a nickel-based superalloy with fine grains showed longer fatigue life than that of a nickel-based superalloy with coarser grains at the same total strain amplitude. Moreover, the smaller the grain size, the greater the proportion of the small fatigue crack stage in the total fatigue life of the nickel-based superalloy GH4169 [15]. Gribbin et al. [16] researched the role of porosity in the direct metal laser sintered Inconel alloy 718 and found that the porosity has less effect on the fatigue life compared to annealing. The fatigue life of the nickel-based 4169 superalloy increased with increasing volume fraction of the δ phase [17]. The γ’ phase is sheared by dislocations at different temperature, resulting in the transition of the fracture mode from crystallographic plane facets fracture (<600 °C) to a mixed fracture mode (600–700 °C) and non-crystallographic fracture (>700 °C) [18]. Qin et al. [19] investigated the influence of stress on the γ″ precipitation behavior in Inconel 718 during aging and found that the γ″ precipitates exhibited homogeneous nucleation, which is strongly promoted by external stress in the early stages of aging.

Hot isostatic pressing is a process that can eliminate internal pores, densify microstructure, and further improve casting quality. Heat treatment can homogenize the original morphology of the microstructure and increase the number of γ″ precipitates [20,21,22]. Qiu et al. [23] found that the ultimate tensile strengths and elongation increased considerably in the nickel-based superalloys, but the yield strengths decreased when the hot isostatic pressing temperature was increased. Chang et al. [24] indicated that, after direct aging, the ductility (elongation > 16%) and yield strength (944 MPa) of Inconel 718 superallloy showed best performance in other different treatments. The polycrystalline nickel-based superalloy had long fatigue cycle life of ~1.5 × 10^4^ cycles since casting pore and acicular δ phases were eliminated by hot isostatic pressing [25]. Liu et al. [26] also found that numerous original cracks were observed in Laves band at the interdendritic regions of micropore, and stress concentration was the dominant factor of microcracks formation in Laves bands through the simulation results. However, extensive studies have been focused on the effects of heat treatment process parameters, laves phase, δ phase, and porosity on mechanical properties and fatigue properties of the nickel-based 718 superalloy [25,26,27,28]. The effects of the γ″ phases on fatigue performance in the nickel-based 718 superalloy have been rarely studied [29]. As a multicomponent and multiphase alloy, the microstructural evolution after HIP and heat treatment becomes fairly complex. Therefore, the effects of the γ″ phases on the fatigue behavior of the nickel-based 718 superalloys after different heat treatments were systematically investigated in this paper.

## 2. Experimental Procedures

### 2.1. Samples Preparation

The materials used were polycrystalline nickel-based 718 superalloys, which were produced in a vacuum induction furnace using investment casting processing under a protection environment of mixed CO_2_ and SF_6_ gas. Melting was introduced into a ceramic shell preheated at 200 °C. Inductively coupled plasma-atomic emission spectrometry (ICP-AES, PerkinElmer, City, US State abbrev., USA) was used to determine the actual composition of the polycrystalline samples. The results are listed in Table 1. Different treatments of standard heat treatment (SHT), hot isostatic pressing + solution treatment + aging (HIP + STA), and hot isostatic pressing + direct aging (HIP + DA) were applied to the as-cast specimens to homogenize the original morphology and eliminate the porosity of the nickel-based 718 superalloys. Table 2 lists the detailed process of the post-treatment in all samples.

### 2.2. Characterization

The phase composition of nickel-based 718 superalloys after treatments of SHT, HIP + STA, and HIP + DA were analyzed by using X-ray diffraction (XRD, Bruker, City, Germany) equipped with a CuK_α1_ radiation source and a step size of 0.02°. For micromorphology observation, all samples were ground and polished with 400^#^–2000^#^ SiC paper and 5–0.5 μm diamond paste successively on the MP-2 metallographic polishing machine. After etched with a mixed solution of 5 g CuCl_2_ + 100 mL C_2_H_5_OH +100 mL HCl for several seconds, an optical microscope (OM, Olympus BX51M, Tokyo, Japan) and scanning electron microscope (SEM, TESAN, Klíčany, Czech Republic) equipped with energy dispersive spectroscopy (EDS) were used to observe the microstructures of all samples. For the analysis of the strengthening precipitates, disc shaped samples with a diameter of 3 mm and thicknesses of ~70 μm were twin-jet electro polished by an electrolyte solution of 5% HClO_4_ and 95% C_2_H_5_OH in a temperature of −20 °C. The dislocation precipitate interactions of all samples were observed by transmission electron microscopy (TEM, JEOL-2100F, Tokyo, Japan).

### 2.3. Fatigue Experiments

The fatigue test samples of high cycle fatigue (HCF) and low cycle fatigue (LCF) were traditional round bars and designed on ASTM E466 and ASTM E606, respectively, as shown in Figure 1. All fatigue tests were carried out by the MTS647 Hydraulic Wedge Grip microcomputer controlled electro-hydraulic servo testing machine with a computer-controlled program to control stress and strain. All the samples of HCF and LCF were machined by wire-electrode cutting and polished with surface roughness of 1.6 and 0.4 μm, respectively. The fracture analysis of the fatigue test samples was performed by SEM and the deformation behaviors of strengthening precipitates in the failure samples were observed by TEM.

HCF experiments of axial tension and compression stress control were carried out at room temperature, where the axial stress was set as 380, 460, 540, and 700 MPa, respectively. HCF experiments were conducted with triangular waves with a stress ratio of R = −1 and loading frequency of = 0.5 Hz. The number of fatigue cycles before the HCF sample failed was recorded. 

LCF experiments of axial tension and compression strain control were carried out at room temperature, where the strain amplitude ε was set as 0.4% and the strain ratio R was set as –1. LCF experiments were conducted with triangular waves with a loading frequency of = 0.5 Hz. Additionally, the stress peak and fatigue cycle before sample failure were monitored and recorded by an extensometer and computer.

## 3. Results and Discussion

### 3.1. Microstructure

Figure 2 shows the XRD patterns of the nickel-based 718 superalloys after different heat treatments. As shown in Figure 2, the as-cast sample consists of γ-Ni, γ’-Ni_3_[Al, Ti], γ″-Ni_3_Nb, Laves-Fe_2_Nb, δ-Ni_3_Nb, and carbide-[Nb, Ti]C. It is noticeable that δ phase (orthorhombic, Ni_3_Nb) has the same chemical compositions with the strengthener γ″ (bct, Ni_3_Nb). After the treatment of SHT, Laves phase decomposition caused the formation of δ and γ″ phase. Therefore, the peak intensity of the δ phase increased and the peak intensity of the Laves phase decreased in the SHT sample. With the treatment of elevated temperature and extreme high pressure, the Laves phase reduced significantly and the γ, γ’, and γ″ phases increased in the HIP + STA and HIP + DA samples. In addition, there were still few δ phases in the HIP + STA and HIP + DA samples. Carbides were present in all samples, which illustrated that the carbide has better thermodynamic stability than other precipitates, as shown in Figure 2.

The optical microstructures of as-cast, SHT, HIP + STA, and HIP + DA nickel-based 718 superalloys are shown in Figure 3. The granular carbide particles and dendritic Laves phases surrounded by black segregation regions are exhibited in the as-cast sample (Figure 3a). It can be seen from Figure 3b that a large amount of Laves phases remained in the interdendritic segregation regions and acicular δ-Ni_3_Nb phases precipitated around the residual segregation regions of the sample after SHT treatment. Although the Laves phase has been eliminated by HIP + STA treatment, relatively few acicular δ-Ni_3_Nb phases precipitated in the interdendritic and along the grain boundaries, as shown in Figure 3c. The further DA after HIP process can form fine network grain boundaries in the sample (Figure 3d). The carbide particles were present in all specimens after different heat treatment, as shown in Figure 3.

Figure 4 shows the SEM images of the HIP + DA nickel-based 718 superalloy. As shown in Figure 4a, many disk-like γ″ phases and cuboidal γ’ phases were dispersed homogeneously in the matrix of the HIP + DA sample. It was worth noting that the fine network grain boundaries in the HIP + DA sample were composed of relatively coarse disk-like γ″ phases with major axes of ~ 100 nm, as shown by the arrow in Figure 4b. As shown in the magnification in Figure 4b, there were some smaller secondary disk-like γ″ phases with major axes of ~50 nm in the matrix, which are smaller than the γ″ phases in the grain boundaries. 

To further identify the crystalline structure of precipitates in the nickel-based 718 superalloys, analysis of the bright-field TEM images of the acicular δ phase and carbide particle, along with the corresponding selected area electron diffraction (SAED) patterns were obtained, as shown in Figure 5. It can be seen that the acicular δ-Ni_3_Nb phase was distributed in the γ matrix of all samples (Figure 5a). Additionally, the angular carbide particle measuring about 4 μm appeared in all samples, as shown in Figure 5b.

Figure 6 shows the TEM dark field images and corresponding SAED pattern of the nickel-based 718 superalloys after different heat treatments. As shown in Figure 6, the morphologies of the γ″ precipitates in all samples were quite similar, which present disk-like microparticles perpendicular to one another. The Image-Pro Plus 6.0 image analyzer) was applied to measure the volume fractions and sizes of the γ″ phases in all samples, and the results were listed in Table 3. All measurements were repeated five times to reduce errors. It can be seen that γ″ phases in the HIP + STA sample have the highest volume fraction (26.4 ± 0.3%) with mean major axes of 23.1 ± 3.7 nm and mean minor axes of 8.4 ± 1.0 nm. Additionally, γ″ phases in the HIP + DA sample present the largest mean major axes of 27.8 ± 3.8 nm and mean minor axes of 8.8 ± 0.8 nm (Figure 6c). The disk-like particles in the nickel-based 718 superalloys were identified to be the γ″ phase, based on the corresponding SAED pattern with an electron beam parallel to the [001] zone, as shown in Figure 6d.

### 3.2. HCF Performance

Figure 7 presents the HCF life of the SHT, HIP + STA, and HIP + DA nickel-based 718 superalloys under different stress levels (380, 460, 540, and 700 MPa). HCF refers to the fatigue caused over 10^5^ to 10^7^ cycles of a material under a cyclic stress less than its yield strength. Generally, the cyclic stress frequency of high cycle fatigue is low, and the strain cyclic amplitude is limited to the elastic range. The fatigue life of the SHT, HIP+ STA, and HIP+ DA samples all decreased with an increase of stress amplitude level and successively increased at the same stress level. Compared to the samples of HIP + STA and SHT, the HIP + DA nickel-based 718 superalloy over different stresses showed the longest fatigue life. Moreover, the fatigue life of the HIP+ DA sample with stress amplitude of 380 MPa was increased to 5.16 × 10^5^ cycles, which was 134.7% higher than that of the SHT sample and 24.2% higher than that of the HIP+ STA sample, as shown in Figure 7.

### 3.3. LCF Performance

Figure 8 shows the LCF life of the SHT, HIP + STA, and HIP + DA nickel-based 718 superalloys. LCF refers to the fatigue failure of materials at a low cycle, which generally does not exceed 10^5^ cycles. Compared to the SHT sample, the fatigue life of the HIP+ STA sample increased by 81%, from 8.35 ×10^3^ to 1.55 × 10^4^ cycles (Figure 8a). Additionally, the HIP + DA sample has the longest fatigue life of 1.70 × 10^4^ cycles, which was 103.8% higher than the SHT sample and 10.2% higher than the HIP+ STA sample. With the increase in cycle times, the stress amplitude of all samples showed a trend of rapid increase and then slow decline, which meant that all samples exhibited initial cycle hardening, short-term cycle saturation, and subsequent cycle softening, as shown in Figure 8b. The cycle hardening times of SHT, HIP + STA and HIP + DA were 10 cycles, 75 cycles, and 1000 cycles, respectively. Compared to other samples, the HIP + DA sample has a longer cycle hardening time and a higher cycle hardening degree.

### 3.4. Fatigue Fracture Analysis

The fatigue fracture surfaces of the nickel-based 718 superalloys are composed of three main zones, including (Ⅰ) fatigue crack initiation zone, (Ⅱ) fatigue propagation zone, and (Ⅲ) final fracture zone [30]. Figure 9 shows the fatigue crack initiation zone of the SHT, HIP + STA, and HIP + DA nickel-based 718 superalloys after a fatigue test of 380 MPa. The fatigue crack initiation zone is the region where the crack is first nucleated, and it usually occupies a small area fraction of the fracture surface compared to the further stages of failure. As shown in Figure 9a, there were network Laves phases in the crack initiation zone of the SHT sample. The fatigue crack initiation zone of the HIP+STA sample exhibited many carbide particles and acicular δ phases, which were at the sub-surface below the main crack surface (Figure 9b). Additionally, there were secondary cracks on the carbide particle, and many slip bands around the carbide were observed, as shown in Figure 9c. Based on the above analysis, the fatigue crack initiation mechanism of samples is generally manifested as follows: (Ⅰ) when the sample contains many Laves phases, the fragile Laves phases become the weakest point of the sample, leading to the initiation of a fatigue crack on the surface of the Laves phase, (Ⅱ) the interfacial strength between the carbide and matrix is weakened by the existence of the acicular δ phase, resulting in a lower interfacial strength than that of the matrix, so the fatigue cracks propagate in the acicular δ phase, and (Ⅲ) the interaction between the resident slip bands and carbide results in the initiation of fatigue cracks at the carbide inclusions near the surface.

Figure 10 shows the fatigue crack propagation zone of the SHT, HIP + STA, and HIP + DA nickel-based 718 superalloys after a fatigue test of 380 MPa. Plenty of uniform, continuous fatigue striations are present in the SHT and HIP + STA nickel-based superalloys (Figure 10a,b). Generally, the number of striations is approximately equal to the number of loading cycles. The distance between the striations becomes wider as the distance from the fatigue initiation zone increases. Due to the obstruction effect of the dislocation of the relatively coarse γ″ phases in the grain boundaries in the HIP + DA sample, the spacing of the fatigue striations is small, and the striations are formed discontinuously, as shown in Figure 10c. It illustrates that the grain boundaries hinder the fatigue crack propagation in the sample.

Figure 11 shows the final fracture zone of the SHT, HIP + STA, and HIP + DA nickel-based 718 superalloys after a fatigue test of 380 MPa at 380 MPa. When the stress exceeds the strength limit of the material, instantaneous fracture will occur. As shown in Figure 11a, the size and percentage of dimples in the final fracture zone in the SHT sample were small and no inclusions appeared in the dimples. Additionally, the fracture morphology was mixed morphology of quasi-cleavage and dimple. Compared to the SHT sample, the final fracture zone of the HIP+STA sample possessed relatively flat dimples (Figure 11b). Figure 11c shows many dimples with different and uneven shapes in the HIP + DA sample. There were bright white tearing edges on the fracture in the HIP + DA sample, which indicated that the deformation process was not uniform, and there was strong plastic deformation.

Figure 12 shows the TEM dark field images of the HIP + DA nickel-based 718 superalloy after a fatigue test of 380 MPa. The γ″ phases with major axes of ~25 nm and minor axes of ~10 nm in the matrix of the HIP + DA sample were cut by dislocation, as shown by the arrow in the Figure 12a. Additionally, Figure 12b shows the relatively coarse γ″ phases with major axes of ~80 nm and minor axes of ~40 nm in the HIP + DA sample. It should be noted that the relative coarse γ″ phases were cut twice by dislocations. The relatively coarse γ″ phases in the grain boundaries increased the dislocation motion resistance, which played an important role in improving the fatigue life. 

## 4. Conclusions

In this study, the effects of γ″ phases on fatigue behavior of nickel-based 718 superalloys with treatments of SHT, HIP + STA, and HIP + DA have been investigated and analyzed in detail. The follow conclusions can be derived from the study:
Under the treatments of the SHT, HIP + STA and HIP + DA, the volume fraction of γ″ phases in the samples increased from 18.5% (SHT) to 26.4% (HIP + STA) and 24.3% (HIP + DA). However, many relatively coarse γ″ phases with major axes of ~80 nm and minor axes of ~40 nm were only observed in the grain boundaries of the HIP + DA sample.With a stress amplitude of 380 MPa, the HIP + DA sample showed the longest HCF life of 5.16 × 10^5^ cycles. Additionally, the HIP + DA sample showed the longest LCF life of 1.70 × 10^4^ cycles, which was higher than the SHT and HIP+ STA samples.Laves phases and carbide inclusions near the surface of the samples were the sources of fatigue fracture. Additionally, the acicular δ phases were the carriers for fatigue crack propagation.The γ″ phases with high content in the matrix and relatively coarse γ″ phases in the grain boundaries of the samples can hinder dislocation movement, which has a great influence on the fatigue life improvement.


## Figures and Tables

**Figure 1 materials-12-03979-f001:**
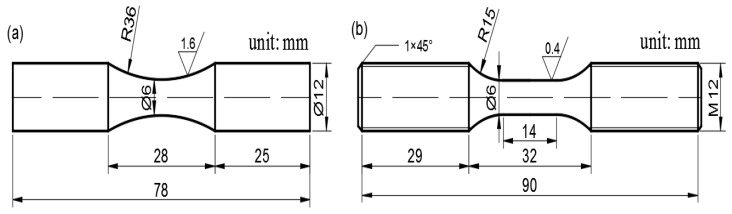
Schematic geometry of fatigue testing specimens: (**a**) high cycle fatigue (ASTM E466); (**b**) low cycle fatigue (ASTM E606).

**Figure 2 materials-12-03979-f002:**
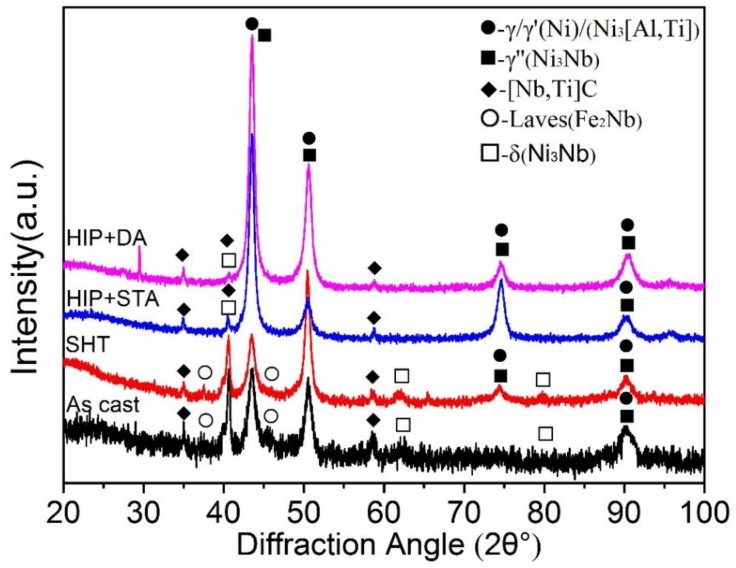
X-ray diffraction patterns of the nickel-based 718 superalloys after different heat treatments.

**Figure 3 materials-12-03979-f003:**
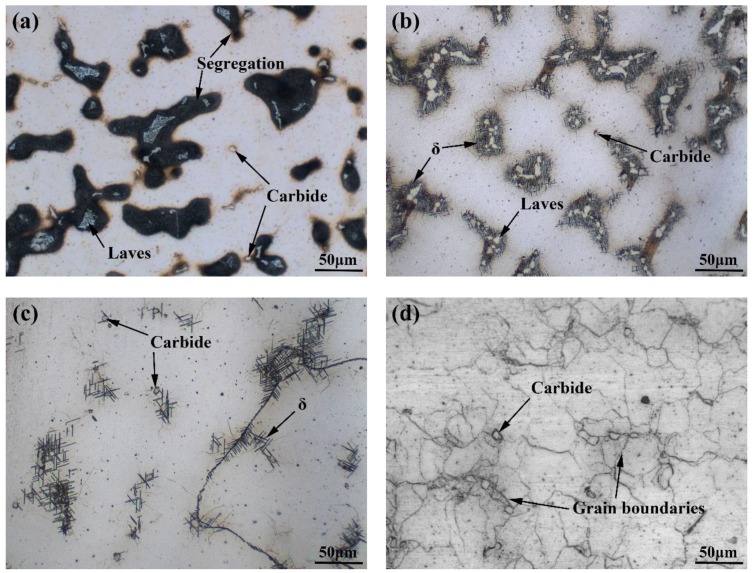
Optical microstructures of nickel-based 718 superalloys after different heat treatments: (**a**) as-cast, (**b**) standard heat treatment (SHT), (**c**) hot isostatic pressing + solution treatment + aging (HIP + STA), (**d**) hot isostatic pressing + direct aging (HIP + DA).

**Figure 4 materials-12-03979-f004:**
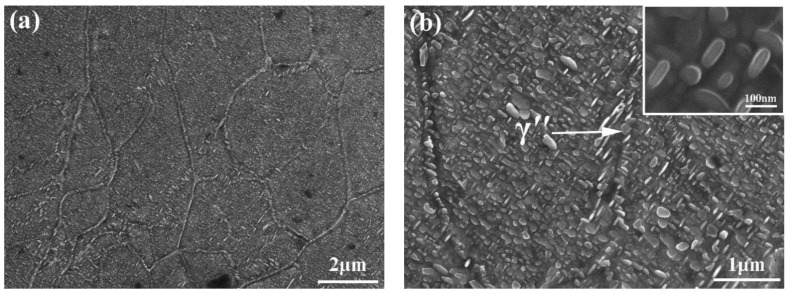
SEM images of the HIP + DA nickel-based 718 superalloy: (**a**) γ″ phases in the interdendritic, (**b**) γ″ phases in the grain boundaries.

**Figure 5 materials-12-03979-f005:**
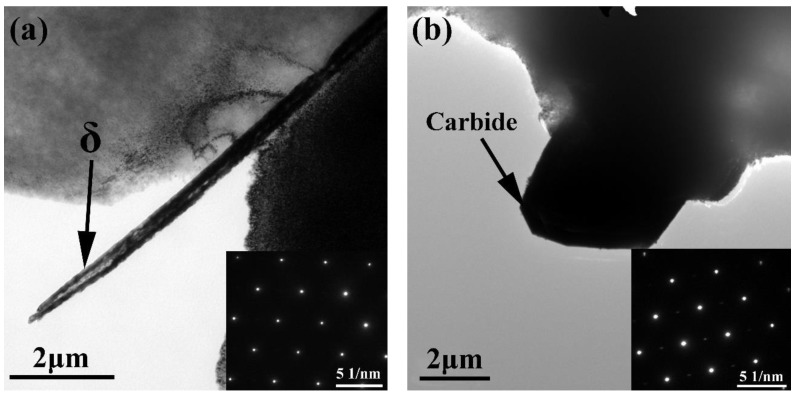
TEM bright field images and selected area electron diffraction patterns of the nickel-based 718 superalloys: (**a**) acicular δ phase, (**b**) angular carbide particle.

**Figure 6 materials-12-03979-f006:**
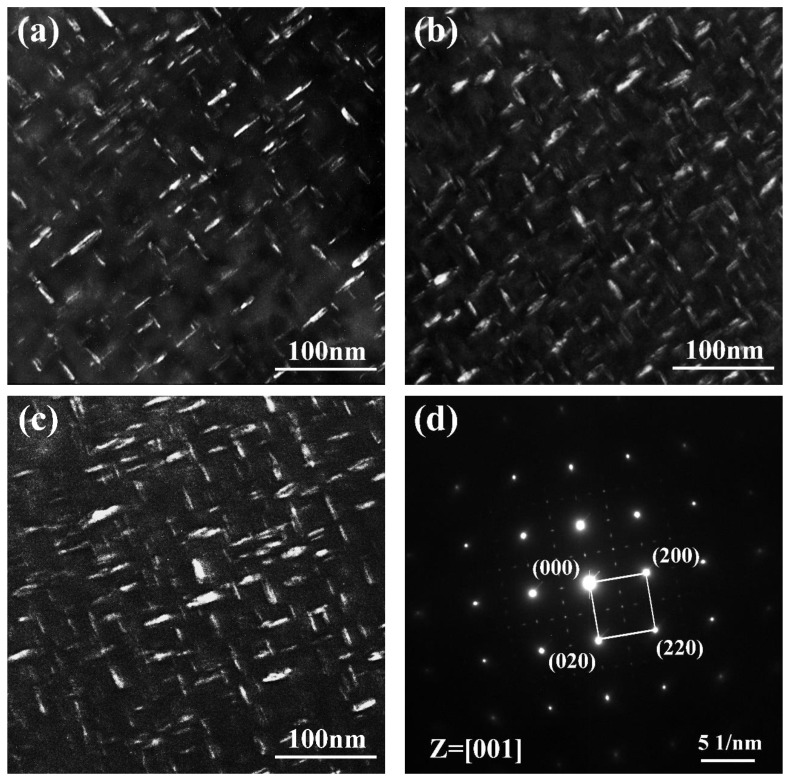
TEM images of the γ″ phase in samples using the (010) and (1/210) reflections in {001} orientation: (**a**) dark field (DF) image of SHT sample; (**b**) DF image of HIP + STA sample; (**c**) DF image of HIP + STA sample; (**d**) SAED pattern of γ″ phase.

**Figure 7 materials-12-03979-f007:**
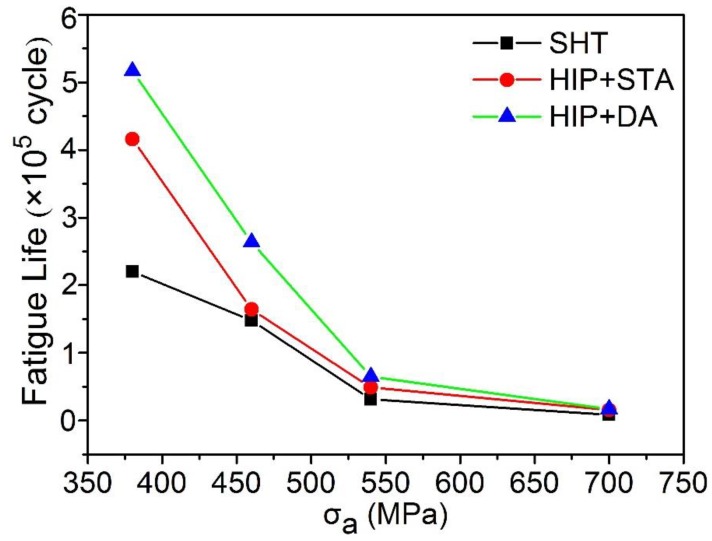
The high cycle fatigue life of the SHT, HIP + STA, and HIP + DA nickel-based 718 superalloys under different stress levels.

**Figure 8 materials-12-03979-f008:**
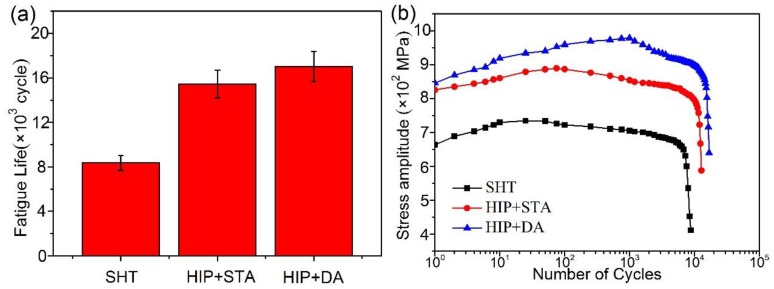
The low cycle fatigue life of the SHT, HIP + STA, and HIP + DA nickel-based 718 superalloys: (**a**) the fatigue life; (**b**) the stress amplitude.

**Figure 9 materials-12-03979-f009:**
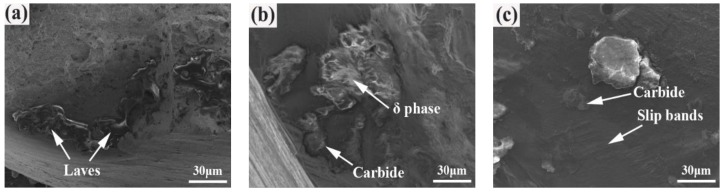
Fatigue crack initiation zone of the nickel-based 718 superalloys after a fatigue test of 380MPa: (**a**) SHT; (**b**) HIP + STA; (**c**) HIP + DA.

**Figure 10 materials-12-03979-f010:**
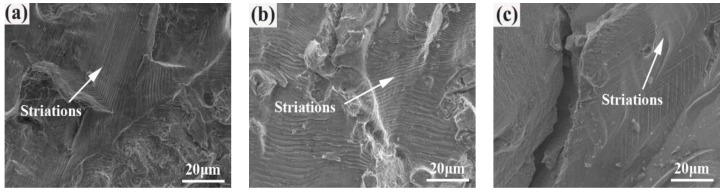
The fatigue crack propagation zone of the nickel-based 718 superalloys after a fatigue test of 380 MPa: (**a**) SHT; (**b**) HIP + STA; (**c**) HIP + DA.

**Figure 11 materials-12-03979-f011:**
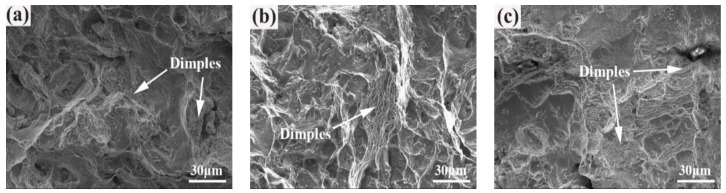
The final fracture zone of the nickel-based 718 superalloys after fatigue test of 380 MPa: (**a**) SHT; (**b**) HIP + STA; (**c**) HIP + DA.

**Figure 12 materials-12-03979-f012:**
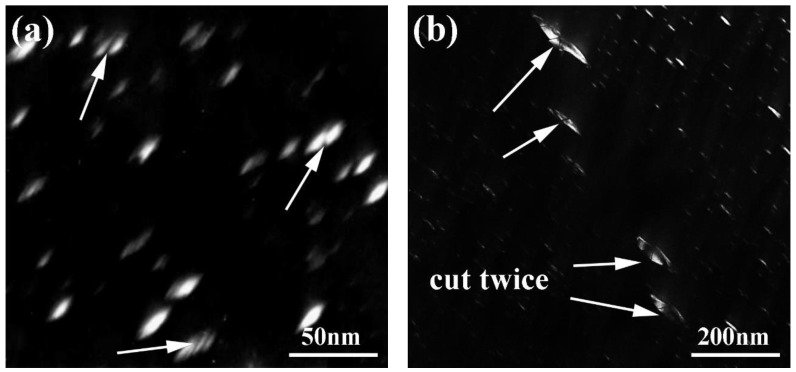
TEM dark field images of the HIP + DA nickel-based 718 superalloy after a fatigue test of 380 MPa: (**a**) small γ″ phases in the matrix, (**b**) relatively coarse γ″ phases.

**Table 1 materials-12-03979-t001:** Actual compositions of nickel-based 718 superalloys (wt.%).

C	Cr	Ni	Co	Mo	Al	Ti	Nb	Fe
0.06	19.43	52.90	0.18	3.15	0.41	1.06	4.36	Bal

**Table 2 materials-12-03979-t002:** The detailed process of post-treatment in nickel-based 718 superalloys.

Designation	Hot Isostatic Pressing	Homogenization	Solution	Aging
As-cast	None	None	None	None
SHT	None	1095 °C/1 h/AC	950 °C/1 h/AC	720 °C/8 h/FC at 55 °C/h620 °C/8 h/AC
HIP+STA	1170 °C/140 MPa/4 h/FC	None	950°C/1h/AC	720 °C/8 h/FC at 55 °C/h620 °C/8 h/AC
HIP+DA	1170 °C/140 MPa/4 h/FC	None	None	720 °C/8h/FC at 55°C/h620 °C 8 h/AC

SHT: standard heat treatment HIP + STA: hot isostatic pressing + solution treatment + agingHIP + DA: hot isostatic pressing + direct aging

**Table 3 materials-12-03979-t003:** Volume fractions and mean sizes of the γ″ phase in the nickel-based 718 superalloys after different heat treatments.

Samples	SHT	HIP + STA	HIP + DA
Mean major axes (nm)	22.6 ± 4.1	23.1 ± 3.7	27.8 ± 3.8
Mean minor axes (nm)	8.1 ± 0.8	8.4 ± 1.0	8.8 ± 0.8
Volume fraction (%)	18.5 ± 0.2	26.4 ± 0.3	24.3 ± 0.1

SHT: standard heat treatmentHIP + STA: hot isostatic pressing + solution treatment + agingHIP + DA: hot isostatic pressing + direct aging

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
