# Peer review of "Effects of the γ″-Ni3Nb Phase on Fatigue Behavior of Nickel-Based 718 Superalloys with Different Heat Treatments"

_materials, 2019, doi:10.3390/ma12233979_

Round 1
Reviewer 1 Report
Review of Effects of the γ″-Ni3Nb phase on fatigue behavior of nickel-based 718 superalloys with different heat treatments
In general the paper is interesting and well written, just a few things that should be improved.
- I think it would be helpful if you write in the figure caption of the Figure 1 the ASTM number.
- Concerning the microstructure, in the XRD could you add some discussion to the obtained results? And the peak assigned to the Laves phase at around 46°, could it be also associated to the delta phase?.
- Concerning the vol. fraction calculation from images of Figure 6: You comment in Figure 4 that there are gamma’’ precipitates of mayor axes of around 100 nm at the grain boundaries. Why we do not see any of those precipitates in the images showed in Figure 6? Accordint to this, is the analyzed area big enough for a real vol. fraction calculation? Can you please comment on this?
- In Figure 12: Why we do not see any gamma’’ precipitates perpendicular to those showd, similar to Figure 6?
Recommendations for orthographic corrections:
- Always leave before a reference à for example “superalloys [7]” instead of “superalloys[7]” (page 1).
- Always leave a space between number and unit à for example “5 g” and not “5g” (page 3).
- Page 3, when you write “polished with 400#-2000# SiC”, shouldn’t you write grit?
- Page 9, “(Figure 8a).” instead of “(Figure 8a.”
Reviewer 2 Report
This paper represents an interesting and up to now missing systematic contribution to coupling of microanalysis of phases and microstructures and mechanical, technologically relevant properties.
The review of state of the art is clear and quite comprehensive. The experimental findings are well documented.
Some major points need to be clarified and improved for publishing:
-) In the introduction, it must be summarized how mechanical properties in IN718 can be improved, i.e. results of Refs 25 to 28 need to be summarized in short and connected to the current gap of knowledge, and how this can be filled by the current research, leading in fact to the point (mentioned) that understanding of the gamma´´ - microstructures - mechanical properties relation is particularly needed.
-) Reference to "rarely studied fatigue behavior" needs to be given.
-) The statement that "many acicular delta-Ni3Nb phases precipitated" in HIP+STA is disagreeing with the one one page before (almost no delta). Clarify.
-) "Nucleation driving force of the gamma´´ phases was reduced and the growth and coarsening...preliminary occured in the later process..." - these interpretations miss a robust base - the differences of mean gamma´´ diameters and lengths are very small among different treatments. In particular, evaluated data require error bars of mean value for more reliable comparisons and interpretations. Further, from which findings of the study can you interprete differences in the nucleation behavior? Clarify.
-) In order to understand the role of heterogeneous gamma´´ precipitation at grain boundaries, the HIP+DA and HIP+STA states need to be compared with the SHT state --.> best would be to extend Fig. 4 for this comparison.
Minor language and style corrections required:
-) Give abbreviations not in the abstract, but in the main part when you introduce the different techniques / states.
-) "among all Inconel 718 superalloys"? "all Ni-base superalloys" instead? Clarify.
-) Do not repeat "Inconel 718 superalloy" so often, it is clear that you research this material. Often not needed to mention again.
-) "corroded" samples --> etched samples.
-) "fracture analysis was characterised" --> fracture analysis was performed, (or similar).
-) "It was worth noting that carbides were present" --> carbides were present.
-) "miniature particles" - re-formulate
-) presented --> present (several times).
